# Knowledge, Attitudes and Practice Regarding Antibiotic Prescription by Medical Interns: A Qualitative Study in Spain

**DOI:** 10.3390/antibiotics12030457

**Published:** 2023-02-24

**Authors:** Germán Molina-Romera, Olalla Vazquez-Cancela, Juan Manuel Vazquez-Lago, Rodrigo Alonso Montes-Villalba, Fátima Roque, Maria Teresa Herdeiro, Adolfo Figueiras

**Affiliations:** 1Preventive Medicine and Public Health Department, University Hospital of Santiago de Compostela, 15706 Santiago de Compostela, Spain; 2Research Unit for Inland Development, Guarda Polytechnic Institute (UDI-IPG), 6300 Guarda, Portugal; 3Health Sciences Research Centre Interior (Centro de Investigação em Ciências da Saúde-CICS/UBI), University of Beira, 6200 Covilhã, Portugal; 4Department of Medical Sciences, iBiMED-Institute of Biomedicine, University of Aveiro, 3810 Aveiro, Portugal; 5Consortium for Biomedical Research in Epidemiology and Public Health (CIBER en Epidemiología y Salud Pública/CIBERESP), 15706 Santiago de Compostela, Spain; 6Health Research Institute of Santiago de Compostela (Instituto de Investigación Sanitaria de Santiago de Compostela—IDIS), University of Santiago de Compostela, 15706 Santiago de Compostela, Spain

**Keywords:** antibiotic-resistance, interns, prescription, qualitative, focus-group techniques

## Abstract

Antibiotic resistance is an issue of growing importance in the public health sphere. Medical interns are of great relevance when it comes to the source of this problem. This study therefore sought to ascertain which factors influence the management of antibiotic therapy by this population, in order to pinpoint the possible causes of misprescribing habits. We conducted a qualitative study based on focus group techniques, with groups consisting of medical interns from the Santiago de Compostela Clinical University Teaching Hospital. Our study identified factors which the participants considered to be determinants of antibiotic use and their relationship with the appearance of resistance. The single most repeated factor was the influence of the attending physician’s judgement; other factors included a high healthcare burden or prescribing inertia. This stage is an opportunity to correct misprescribing habits, by implementing educational interventions aimed at modifying the identified factors.

## 1. Introduction

Infections due to multiresistant bacteria are a serious public health problem [1], due to the morbidity, mortality and financial burden that they generate [2]. The problem of resistance is so serious that cases in which there are no antibiotics to treat certain hospital infections are becoming increasingly common [3]. Similarly, it is thought that if this trend continues, it might not be possible for some medical acts to be performed, e.g., certain surgical operations, or even transplants [4,5]. One of the main causes of multiresistance is inappropriate use of antibiotics [6,7,8]. This problem is especially noticeable in developing countries. Spain is one of the countries in Europe with the highest consumption of antibiotics [9,10].

In this respect, a meta-analysis has shown that less than half of all hospitalised patients receive optimal antibiotic treatment in accordance with prescription guidelines [11]. This has been exacerbated by the recent COVID-19 pandemic [12]. Hospital physicians play a key role in such inappropriate prescribing. Yet, the reasons that give rise to these habits are not well known. One of the most decisive factors might be physicians’ knowledge and attitudes, many of which are acquired during their time as medical interns. This stage is of key importance in the training of physicians, since it is when they become specialists and acquire the necessary skills, habits and behaviour patterns to perform their professional duties, including the prescribing of antibiotics [13]. A systematic review of final-year medical students pointed to lack of knowledge and difficulties in accessing clinical practice guidelines as the main factors in inappropriate prescribing [14].

In contrast to quantitative methods, qualitative methods in general and focus groups in particular, have the great advantage of enabling all the dimensions of the problem to be detected [15]. Hence, they make it possible to identify knowledge, attitudes, perceptions and beliefs (KAPB), as well as external factors, such as the interns’ relationship with attending physicians, aspects about which the participants themselves are not aware until they emerge in the course of the focus group discussion [16].

Accordingly, this study sought to examine, from a qualitative standpoint, the factors that influence antibiotic use among the medical intern population, with the focus on their KAPB and sources of information used. Identifying these factors would make it possible to design antibiotic-prescribing training strategies at this crucial stage in the training of future health professionals.

## 2. Results

Seven focus groups were set up and held. Each consisted of four to six participants, amounting to a total of 35 medical interns from the Santiago de Compostela Clinical Hospital (Table 1).

After analysis of all the recordings, the main factors cited by the interns as determinants of antibiotic use and its relationship with appearance of resistance, were identified (Table 2).

### 2.1. Factors

The interns identified different items that influenced antibiotic use (Table 3). Some of these were exclusively perceived as factors external to their own routine practice. All the groups analysed agreed that the most decisive and influential factor was the judgement of the attending physician. 

#### 2.1.1. Knowledge

Own knowledge

No important gaps in knowledge were found among the interns, when it came to assessing the need or lack of need to prescribe an antibiotic (Appendix A/1.1/a/FG1M1). They showed themselves to be knowledgeable about and critical of the misuse of certain antibiotics—mainly in the context of upper respiratory tract and urinary infections—despite the fact that these drugs were not indicated (Appendix A/1.1/a/FG2M2).

b.Knowledge among the general population

The interns felt that the general public did not have sufficient knowledge of antibiotics and that this revealed the need for better health education. They were of the opinion that patients considered antibiotics to be the most appropriate treatment for any type of infectious disease or symptoms related to an infection (Appendix A/1.1/b/FG3W1).

#### 2.1.2. Healthcare Burden

The interns regarded the time allotted to attend to each patient, as well as the number of patients, as influential factors in the decision to prescribe an antibiotic (Appendix A/1.2/FG5W5).

In some focus groups, the number of hours worked was identified as a cause of misprescription of antibiotics (Appendix A/1.2/FG2M2).

One intern rejected the idea of time of care, and suggested indifference or associated lack of interest as an element that influenced prescribing. (Appendix A/1.2/FG6M3).

#### 2.1.3. Inertia

The interns had the perception that there were antibiotic prescribing attitudes which were based more on previous experience and routine practice than on available scientific evidence. This attitude was particularly associated with the most common infections and was attributed to prescriber convenience (Appendix A/1.3/FG3W1, FG5W5, FG6M1).

#### 2.1.4. Pharmacological Characteristics

The interns reported the use of certain antibiotics instead of others, with the choice being based more on convenience of the dosage than on evidence-based indications (Appendix A/1.4/FG1W1). Side-effects were also identified as decisive factors in the choice of antibiotic (Appendix A/1.4/FG3M1).

In two focus groups, the form of presentation of antibiotics was described as being a limiting factor for their correct use. Drug presentations with a higher number of doses than those required by the most frequent indications for treatment, tended to lead to excess availability of the antibiotic for the patient, something that in turn favoured inappropriate use. (Appendix A/1.4/FG6M1).

#### 2.1.5. Patient Pressure

Pressure exerted by patients in emergency services and out-of-hospital visits to be treated with antibiotics was identified as a decisive element in the prescription of these drugs.

In many cases, such demands were considered to influence the prescriber’s decision, in acceding to the patient’s request and prescribing an antibiotic where it is not indicated. Furthermore, this is done to avoid confrontation with the patient (Appendix A/1.5/FG6W3).

In one focus group, a participant had previously worked in the private health sector. From his standpoint, he described the attitude of patients who demand a specific treatment which they consider appropriate, simply because they are paying directly for the visit. In turn, there is also the attitude of private health sector physicians who, for the same reason, tend to accede to such demands more frequently than do their public health colleagues (Appendix A/1.5/FG2W3).

#### 2.1.6. Complacency towards the Patient

Participants acknowledged an attitude of complacency when it came to treating patients. This leads to inappropriate use of complementary tests and treatments. It is a matter of physicians trying to provide what they think their patients want, in order to meet their expectations. The interns had the perception that this problem was more frequent in private medicine (Appendix A/1.6/FG6M1).

#### 2.1.7. Complacency towards other Physicians

Interns also viewed the variability of judgement among attending physicians as a limitation, when it came to choosing the appropriate treatment. They described the existence of a behaviour of complacency, as a result of which the attending physicians themselves will not change a colleague’s decision, despite showing themselves to be in disagreement with it and actually questioning it (Appendix A/1.7/FG2M2, FG4W1).

#### 2.1.8. Fear

In many cases the prescribing of antibiotics is considered necessary, without there being a definitive diagnosis or aetiology of the defined disease. There is an attitude of fear at the prospect of a poor disease course developing, in light of certain signs and symptoms. This situation leads to antibiotic treatment being prescribed “just in case” (Appendix A/1.8/FG5W5).

#### 2.1.9. Attending Physician’s Judgement

In their own practice, interns considered their attending physician’s judgement to be an extremely decisive and influential factor when it came to prescribing an antibiotic. They saw such judgement as being the end result of the conjunction of the above-mentioned factors (Appendix A/1.9/FG1W2).

In two focus groups, two interns highlighted the existence of a theory-practice gap between knowledge acquired at university and its application in routine clinical practice (Appendix A/1.9/FG3M3).

In one focus group made up of interns from a more advanced stage of the training period, the attending physician’s judgement was not identified as being an important influence over their own clinical opinion. Nevertheless, prescriptions which had been issued by attending physicians and which they considered incorrect, continued to be left untouched (Appendix A/1.9/FG7M1).

#### 2.1.10. External Responsibility

The interns felt there was multifactorial responsibility for the appearance of multiresistance. The principal actor identified was the primary care physician and, to a lesser extent, the specialist (Appendix A/1.10/FG1M1).

Furthermore, they also regarded the dispensing of antibiotics without prescription by pharmacies as playing an important part in the generation of multiresistance (Appendix A/1.10/FG4W3).

In some focus groups the veterinary industry was identified as the agent responsible (Appendix A/1.10/FG3W1).

### 2.2. Needs

#### 2.2.1. Tests

In all the focus groups, the interns agreed on clinical signs which, together with data yielded by complementary and analytical tests, influenced the decision to prescribe an antibiotic.

In all the groups, great importance was attached to results obtained from blood and urine analysis tests. Specifically, they judged the appearance of leukocytosis or leukocyturia to be a decisive clinical sign for treating patients with antibiotics (Appendix A/2.1/FG5W2).

#### 2.2.2. Training

The sources used by physicians for updating and training purposes were identified as important factors in the prescription of antibiotics. While these essentially consist of clinical guidelines and protocols, they also include hospital antibiotics committees.

The interns said that neither clinical guidelines nor protocols were frequently used. They saw these as being inaccessible, and even went so far as to question their usefulness, with the result that they tended to prescribe without challenging the attending physician’s opinion (Appendix A/2.2/FG1W1).

Antibiotics committees were questioned in the majority of cases. Nonetheless, the interns argued that they ought to be an important presence and felt that they would be of great utility to healthcare staff (Appendix A/2.2/FG2W1).

They expressed the need for more specific training of health professionals, and the drawing-up of clinical guidelines, as well as accessible and updated protocols (Appendix A/2.2/FG5W1).

They also stressed the importance of health education for the general public. Such education should be imparted by institutions through awareness-raising campaigns, and by the healthcare institutions themselves (Appendix A/2.2/FG3W1), (Appendix A/2.2/FG3W3).

## 3. Discussion

Our study shows that, in their daily practice, interns acknowledge misprescribing antibiotics because they follow the attending physicians’ advice, and not checking clinical practice guidelines for updates because the latter are inaccessible. These situations unequivocally lead to antibiotic misprescribing habits being perpetuated and suggest that interventions to improve antibiotic prescribing should be targeted at breaking this chain.

### 3.1. Factors

This study made it possible to identify the factors which, in the interns’ opinion, influence prescribing. These results are in line with previous studies conducted on medical practitioners—attending physicians and interns alike—active in both hospital and primary care. Moreover, among other factors, these studies identified the patients themselves as being responsible for the generation of multiresistance due to misuse/abuse of antibiotics [16,17,18].

### 3.2. Hierarchy and Professional Deference

One of the most relevant conclusions of our study is the acquisition of practice in the use of antibiotics, based on the judgement of the attending physician. Despite being practitioners possessed of the same prescribing capability and adequate knowledge, they do not feel confident enough to challenge the judgement of their attending physicians, even though they are aware that this is occasionally incorrect. As a result, they ultimately prescribe what the attending physician considers best [19,20].

It is in this way that the misprescribing habits of attending physicians are passed on to interns. At some point in the future, these medical interns will in turn be in charge of training new specialists from the position of attending physician. All this perpetuates attitudes and bad praxis in antibiotic use. Moreover, bearing in mind the fact that interns may well start from better, more up-to-date knowledge, this hierarchical relationship means that an opportunity of two-way training is lost, whereby attending physicians would otherwise obtain updated knowledge from their interns and the latter would take advantage of the experience of their better qualified seniors [20,21]. Several interventions in primary care physicians have been shown to reduce antibiotic prescribing. These could be even more beneficial in this group [22].

The junior physicians in our study also describe a behaviour of reticence to change a prescription made by a colleague or express a different opinion with respect to the use of antibiotics [20,23]. In this respect, on observing these types of attitudes, medical interns are going to show even greater reluctance when it comes to expressing their opinions about inappropriate prescriptions issued by physicians of higher professional standing.

### 3.3. Training

Another problem identified in our study is that medical interns regard sources of updating, such as clinical guidelines and protocols, as being inaccessible. This is something that has also been seen in other similar studies conducted in this region and abroad [24,25,26].

Implementation of training programmes entails an improvement in prescribing, by bringing it in line with the information set out in clinical guidelines [27]. At this point, it seems particularly relevant to provide training in antibiotic prescription from the faculty, giving it the weight it has in terms of a Public Health problem [14].

### 3.4. Perspectives and Implications for Health Policy and Planning

After identifying interns’ perceptions regarding the appearance of multiresistance, its causes and determinants, there was agreement on the fact that this is a problem of great relevance and timeliness, whose causality is multifactorial, and different solutions were proposed. However, our study shows that medical interns show perspectives halfway between those published in primary care physicians and the general population, as they are still in their training period [28,29,30].

In terms of responsibility, it was identified that physicians do not accept their role as part of the problem in the development of antimicrobial resistance. This concept of external responsibility was identified in other groups involved in the use of antibiotics, such as pharmacists and veterinarians [18,28,31].

Given that interns admit to being well informed about the prescribing of antibiotics but do not apply this knowledge, due to the prescribing habits imposed by the attending physicians who tutor them, educational interventions to improve antibiotic prescribing should be targeted at establishing a two-way relationship between attending physicians and interns, and a system of learning based on training in which the tutor both involves the intern and makes him/her participate in decisions from the outset [32]. In addition, antibiotic use policies and stewardship programmes can positively influence these professionals just as they are effective in other populations [33,34].

### 3.5. Strengths and Limitations

This study has the strengths and limitations specific to qualitative methodology. The main limitation lies in the number of participants studied, which may not be representative of the total number of interns enrolled in all the medical specialisations. At all events, with the number of interns that were studied, saturation of the information contributed by them was reached, an important criterion of quality in this type of study using our methodology.

One strength of the study is the heterogeneity of its sample, which makes it possible to obtain in-depth information about the use of antibiotics at different levels and from different standpoints.

For the purposes of this study, the COREQ checklist criteria [35] were applied to evaluate the quality of qualitative methodology (see Appendix A).

## 4. Materials and Methods

### 4.1. Settings

In Spain, the medical internship training system (médico interno residente/MIR) is a stage in medical training, in which candidates who pass a selective examination undergo a specific training programme for each medical specialisation, lasting 4 to 5 years. During this period, the interns are rotated through the different hospital and out-of-hospital services, including the emergency service, which are relevant for their respective medical specialisations [36].

### 4.2. Study Design

We conducted a qualitative study using the focus-group method, regarded as the methodology best suited to analysing different experiences and attitudes from the standpoint of those involved. Our study targeted subjective opinions held by the medical community about antibiotic use, seeking a “point of saturation of information” at which no further new ideas would emerge. Valid and reliable results can be obtained in this way, ensuring a fully integrated approach to all the dimensions of the problem [37,38].

The script used during the sessions was drawn up by reference to previous studies based on focus groups of medical specialists, pharmacists and patients [18,29,39].

### 4.3. Study Population

Our target study population was made up of medical interns from Santiago de Compostela Clinical University Teaching Hospital. The participants were recruited via social media, by telephone and through key informants (heads of the respective hospital departments), using convenience sampling, the method of selection most appropriate for the focus group technique [38]. The participants were briefed on the nature of the study and the research objectives: all those invited agreed to participate in the study. The researchers were introduced to the participants and informed of their background, interests and objectives.

### 4.4. Procedure

The focus groups met in a lecture room belonging to the Santiago de Compostela Clinical University Teaching Hospital, and the sessions were moderated by two researchers, one who was acting as session leader and the other as moderator, avoiding parallel conversations or deviations from the established script. Notes were taken during the sessions to prevent subsequent confounding.

The sessions were taped with an iOS-based recording application, using two devices placed in the centre of the group to ensure quality for transcription purposes. The sessions lasted for approximately 35 to 60 min each and were moderated by a researcher (GMR, OVC or RAMV). Focus group sessions were held until a point was reached at which no new ideas were emerging (saturation of information), with this being applied as a criterion of study validity, i.e., once saturation is achieved, study quality is not enhanced by adding more units of information [37]. The sessions were transcribed by one researcher (GMR), with a second researcher (OVC) tasked with ascertaining and correcting possible errors by mutual agreement. Participants were coded by gender range (“M” for men, “W” for women), and each group was identified with a serial number (FG1, FG2, FG3, etc.). The transcriptions were handed to the participants to detect possible disagreements or make corrections.

### 4.5. Analysis

The transcriptions were separately analysed by two researchers, a man and a woman, both medical interns in the Preventive Medicine and Public Health Department of Santiago de Compostela University (Galicia) (GMR, OVC), in order to reduce any risk of research bias [40].

We used a thematic and discourse analysis of the data, with it being discussed by all authors.

The ideas extracted were then associated with concepts and codified, and new hypotheses were established according to the Constructivist Grounded Theory method [41,42,43]. Differences in interpretation among the researchers were debated and resolved by consensus. No computer software programme was used to process the data.

### 4.6. Ethical Considerations

The study was evaluated and approved by the Santiago-Lugo Research Ethics Committee. The participants were informed of the purpose of the study and the plan to record and transcribe the sessions, maintaining anonymity in the analysis of the recordings. All participants were briefed and gave their verbal consent to participate in the study.

## 5. Conclusions

Improvement in antibiotics prescribing calls for complementary approaches from different settings. There seems to be a culture of antibiotic use and abuse. This article shows that this habit is acquired at the earliest stages of a doctor’s training. However, these are modifiable factors. These findings may well be of great utility when it comes to designing more direct, higher impact campaigns aimed at preventing the perpetuation of this inappropriate prescribing culture.

## Figures and Tables

**Table 1 antibiotics-12-00457-t001:** Focus group characteristics.

	Participants	Men/Women	Medical Specialisation of Residency
FG1	5	2/3	ICU, Traumatology, Cardiology, General Surgery and Anaesthesia
FG2	5	2/3	Gynaecology, Neurosurgery, Psychiatry, Internal Medicine and Nephrology
FG3	4	1/3	Neurology, Internal Medicine, Ear, Nose and Throat/ENT (Otorhinolaryngology) and Oncology
FG4	5	1/4	Paediatrics
FG5	5	0/5	Gynaecology
FG6	5	2/3	ENT
FG7	6	4/2	Internal Medicine, Vascular Surgery, Neurosurgery, General Surgery and Oncology

FG: Focus group. ICU: Intensive Care Unit.

**Table 2 antibiotics-12-00457-t002:** Determining factors of antibiotic use.

Blocks	Coding	
1. Factors	1.1 Knowledge	Own. General population.
	1.2 Healthcare burden	Time available per patient. Number of hours worked. Number of patients.
	1.3 Inertia	Prescribing based on previous experience.
	1.4 Pharmacological characteristics	Prescribing based on convenience of dosage and adverse effects rather than indication of the antibiotic.
	1.5 Patient pressure	Direct patient pressure to obtain antibiotics.
	1.6 Complacency towards the patient	Complacency towards the patient to fulfil his/her expectations.
	1.7 Complacency towards other physicians	Not changing the prescribing decisions taken by other health professionals.
	1.8 Fear	Not having patient follow-up and not knowing how he/she will progress. Not having an accurate diagnosis and prescribing antibiotics without clear indication.
	1.9 Judgement of attending physician	Prescribing based on indications of the physician in charge of training, despite the possibility of these being erroneous.
	1.10 External responsibility	Responsibility of other sectors apart from medical interns, such as pharmacies, primary care physicians, or the patients themselves.
2. Needs	2.1 Training	Updated protocols and guidelines. Health education for the population.
	2.2 Tests	Improvement in the use of complementary tests.

**Table 3 antibiotics-12-00457-t003:** Results of the FG sessions.

FG1	FG2	FG3	FG4	Factors	FG5	FG6	FG7
				Lack of knowledge			
✓	✓	✓	✓	Healthcare burden	✓		✓
✓	✓	✓	✓	Inertia	✓	✓	✓
✓	✓	✓	✓	Pharmacological characteristics	✓	✓	
✓	✓	✓	✓	Patient pressure	✓	✓	
	✓	✓	✓	Complacency towards the patient	✓	✓	
			✓	Complacency towards other physicians			✓
✓	✓	✓	✓	Fear	✓	✓	✓
✓	✓	✓	✓	Judgement of attending physician	✓	✓	✓
✓	✓	✓	✓	External responsibility	✓	✓	✓
				**Needs**			
✓	✓	✓	✓	Training	✓	✓	✓
✓	✓	✓	✓	Tests			

## Data Availability

The transcriptions used and/or analysed during the current study are available from the corresponding author on reasonable request.

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
