# Peer review of "Knowledge, Attitudes and Practice Regarding Antibiotic Prescription by Medical Interns: A Qualitative Study in Spain"

_antibiotics, 2023, doi:10.3390/antibiotics12030457_

Round 1

Reviewer 1 Report

Well done!

Reviewer 2 Report

Studies involving antimicrobial use are important. Results from these studies are essential in assisting policy-makers to design evidence-based healthcare interventions. Although the focus of this study is not new, the study locations being in Spain means this study can make a contribution to the literature and provide insight into what is a role of interns in improving AMU. This paper has a potential to be accepted, but some central points must be explained before going forward.
I’d suggest to change title because it is bit confusing; replace term determinants which is more quantitative 
The authors are requested to collect and add more information from the papers/literature that are published in 2020-2022 in intro and discussion. The article lacks in this area.
 Overall method is very weak and I'd suggest authors to mention all parameters given in COREQ-32 checklist (https://academic.oup.com/intqhc/article/19/6/349/1791966).
Globally I suggest you write your article towards the international readership of your targeted journal. Further consideration of the broader context and how the findings help inform policy to more clearly discuss the implications of the work.
Almost entire conclusion is vague. It needs to be clearer and more precise. 

Reviewer 3 Report

The manuscript, “Determinants of antibiotic prescribing by medical interns: a qualitative focus”, 

The authors use a survey group format to qualitatively examine the knowledge, attitudes, perceptions, and beliefs among medical interns to find the prevalent cause(s) of inappropriate subscription of antibiotics. As antibiotic resistance continues to be a serious problem it is reasoned that interns should learn early in their medical careers the proper course for addressing mismanagement of antibiotics. The manuscript suggests targeting the key causative agents of antibiotic prescription mismanagement at the intern level through training programs and physician-intern communication. 

Overall, the background, goals, and methods are clearly outlined in the manuscript. The methodology is qualitative in nature and the authors address both the pros and cons of this approach. While interpretation of qualitative data is subjective and often times open-ended, the authors describe their method for reaching consensus of interpretation of survey data and cite several articles to substantiate their findings . While such surveys on this topic are not completely novel (similar studies are cited within the manuscript itself) the current work focuses on early career medical professionals (interns) where there is seemingly less data available. This study would likely benefit the physician-intern mentoring process in terms of opening up lines of communication between mentor and mentee. This is especially since the study notes the hesitation of medical interns to challenge attending physicians despite contrary opinions or knowledge. 

I am puzzled by one particular statement in the manuscript, because it is affirmed strongly in the narrative but doesn’t seem to agree with the data presented in Table 3. Lines 73-74 state:

“the groups analysed agreed that the most decisive and influential factor was the judgement of the attending physician.”

However, Table 3 shows that only 4/7 of the study groups consider the category of “Judgement of attending physician” as important. There are several categories in Table 3 that unanimously agreed upon the importance of a factor such as “Inertia” and “Fear”. If the authors could explain the statement of lines 73-74 as it relates to Table 3 it would be appreciated. 

Round 2

Reviewer 2 Report

I endorse this paper for publication